# SCOPE-MIA: Scale-Consistent Partial Differential Equation-Optimized Encoding in 3D Medical Imaging Analysis

## Abstract

Medical scans such as Computed Tomography (CT) or Magnetic Resonance Imaging (MRI) are inherently 3D, capturing rich volumetric information about patient anatomy and pathology. Analyzing this data requires models sensitive to both large-scale anatomical structures and fine-grained textural details. While traditional handcrafted radiomics features often miss subtle multi-scale relationships, standard 3D Convolutional Neural Networks (CNNs) learn data-driven filters without explicit mechanisms to disentangle structural and textural information. This entanglement can limit model interpretability and robustness. We propose SCOPE-MIA, a 3D framework driven by Partial Differential Equations (PDEs) that explicitly decomposes learned features into distinct anatomical and textural components. By embedding tailored PDE constraints - such as structure-enhancing diffusion for anatomy and detail-preserving flows for texture - into a modern 3D CNN architecture, we promote the learning of robust, scale-explicit, and disentangled representations. Our system processes volumetric data in tractable subvolumes for efficient training, while a sliding window approach during inference recovers the full 3D context. We present a unified mathematical treatment connecting PDE theory to our dual-pathway architectural design and discuss the advantages of this decomposition for clinical applications. Extensive experimental validations demonstrate that our method significantly outperforms the clinical gold-standard radiomics pipeline in challenging cancer imaging tasks, showing its potential to advance tumor characterization and biomarker discovery.

## 1 Introduction

Medical scans such as Computed Tomography (CT), Magnetic Resonance Imaging (MRI), or Positron Emission Tomography (PET) are intrinsically 3D, capturing volumetric data of anatomical structures. In parallel, clinical annotations (e.g., tumor segmentations, disease states) are also specified in three dimensions. *Radiomics* seeks to convert such 3D images into quantitative descriptors that can be used for prognosis, diagnosis, or treatment planning Lambin et al. (2012); Van Griethuysen et al. (2017a). Traditional handcrafted radiomics features, while interpretable, often struggle to capture the complex interplay between large-scale morphology and subtle textural patterns present in medical images. For instance, comprehensively characterizing diverse tumor types and their surrounding anatomy using full 3D structural-textural information is crucial, potentially linking imaging features to molecular-level properties like tumor genotype or immune status; however, this remains a challenge. Deep learning approaches, particularly 3D Convolutional Neural Networks (3D CNNs), learn representations directly from volumetric data Litjens et al. (2017), offering a more powerful and flexible alternative. However, standard CNNs typically learn filters that mix features across different scales and types (e.g., smooth boundaries vs. intricate textures) within the same feature maps. This entanglement can hinder interpretability and potentially compromise robustness, as the network lacks explicit mechanisms to treat structural and textural information differently according to their inherent characteristics.

Inspired by classical scale-space theory Witkin (1984); Koenderink (1984); Perona & Malik (1990), which uses Partial Differential Equations (PDEs) (like the heat equation) to analyze images at multiple scales, we propose embedding PDE constraints within a 3D CNN. Our goal is not just to achieve

multi-scale representation, but to leverage PDEs to enforce a principled *separation* of learned features into two complementary streams: one capturing *anatomical* (predominantly low-frequency, structural) information, and another capturing *textural* (higher-frequency, detailed) information. We hypothesize that this explicit decomposition, enforced by tailored PDE dynamics for each stream, will lead to more robust and interpretable radiomic embeddings.

Our method, SCOPE-MIA (Scale-Consistent PDE-Optimized Encoding for Medical Imaging Analysis), implements this concept within a modern 3D CNN backbone Liu et al. (2022). We introduce specialized "PDE Blocks" that operate on the network's latent feature maps. These blocks are designed to simulate specific PDE evolutions: one branch enforces smoothing appropriate for anatomical structures, while the other uses different dynamics to preserve or even enhance textural details. This dual-pathway approach allows the network to learn disentangled representations, guided by the mathematical properties of PDEs. Furthermore, we address the practical challenge of processing large 3D volumes. Training deep models on full scans is often infeasible due to GPU memory limitations. Therefore, SCOPE-MIA adopts a subvolume-based training strategy, processing stacks of $k$ consecutive slices. The PDE constraints play a crucial role here, helping to maintain scale consistency and meaningful feature extraction even when the network only observes a partial volume at a time. During inference, a sliding window approach reconstructs the full 3D context, aggregating predictions from overlapping subvolumes.

This paper makes the following contributions: (1) We propose a novel 3D CNN framework, SCOPE-MIA, that integrates PDE constraints to explicitly decompose learned features into anatomical and textural streams. (2) We derive the PDE formulations tailored for this task, incorporating mechanisms like entropy-adaptive diffusion and information-theoretic attention. (3) We detail an architecture incorporating "PDE Blocks" that implement these dual dynamics within a modern 3D CNN backbone. (4) We demonstrate that this approach integrates seamlessly with a subvolume-based training strategy, and we validate its superior performance against the established clinical gold-standard pipeline across multiple, diverse cancer imaging datasets.

In summary, SCOPE-MIA addresses the lack of principled scale management and feature disentanglement in standard 3D CNNs by embedding PDE constraints. This, combined with a practical subvolume processing strategy, aims to deliver robust, interpretable, and scale-consistent features for demanding clinical applications, enhancing the characterization of tumors and surrounding anatomy.

## 2 RELATED WORK

### 2.1 THE PARADIGM OF HANDCRAFTED RADIOMICS

For decades, quantitative medical image analysis relied on manually engineered features to characterize lesions and surrounding anatomy Lambin et al. (2012); Van Griethuysen et al. (2017a). This paradigm, now often termed "classical radiomics," involves designing a large set of mathematical descriptors to quantify imaging data. These features range from first-order intensity statistics and shape-based measures (e.g., compactness, sphericity) to complex second-order textural features like the gray-level co-occurrence matrix (GLCM), which captures local patterns of voxel intensities Avanzo et al. (2020); Vallières et al. (2015). Classical computer vision descriptors such as the Scale-Invariant Feature Transform (SIFT) and Histograms of Oriented Gradients (HOG) were also adapted for medical tasks Lowe (2004); Dalal & Triggs (2005); Mohanaiah et al. (2013). These handcrafted features are typically fed into machine learning classifiers like support vector machines or random forests for tasks such as tumor classification Subashini et al. (2010); Zhang et al. (2011). To promote standardization and reproducibility, initiatives like the Image Biomarker Standardisation Initiative (IBSI) were formed, leading to robust software packages like PyRadiomics Van Griethuysen et al. (2017b), which has become a gold-standard benchmark. However, the pre-defined nature of these features limits their expressive power, motivating a shift toward data-driven feature learning.

### 2.2 ADVANCES IN 3D DEEP LEARNING FOR MEDICAL IMAGING

The deep learning revolution, catalyzed by the success of Convolutional Neural Networks (CNNs) like AlexNet Krizhevsky et al. (2012) on large-scale image recognition tasks, offered a powerful alternative to manual feature engineering. Architectures such as VGG, ResNet, and ConvNeXT demonstrated the ability to learn hierarchical features directly from data Simonyan & Zisserman

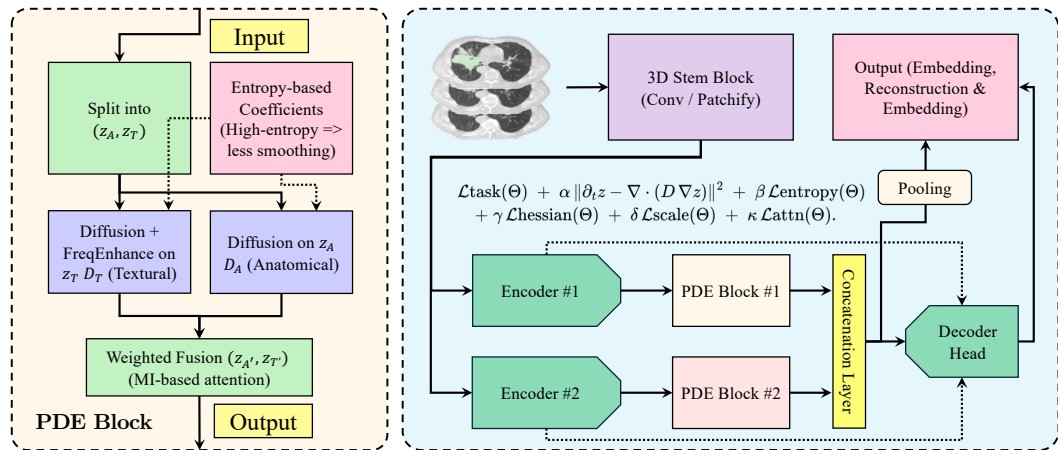

Figure 1: Visualization of the proposed SCOPE-MIA model. Left: Detail of a PDE Block, showing the input feature map $\mathbf{f}$ being split into anatomical ($\mathbf{f}_A$) and textural ($\mathbf{f}_T$) components, processed by parallel PDE-inspired update steps (implementing Eq. 1 and 2 via finite differences), potentially modulated by entropy/attention, and then recombined into the output $\mathbf{f}'$. Right: Schematic architecture of our model where we are incorporating these PDE Blocks at various stages.

(2014); He et al. (2016); Liu et al. (2022). In medical imaging, early applications often involved using 2D CNNs for slice-by-slice analysis of volumetric scans Gour et al. (2020); Heidari et al. (2020); Walsh et al. (2018). However, this approach discards crucial inter-slice contextual information. Recognizing this limitation, the field rapidly moved towards fully 3D architectures. Early work in general computer vision demonstrated the feasibility of 3D CNNs for tasks like voxelized object recognition Maturana & Scherer (2015) and video analysis Tran et al. (2015). In the medical domain, this led to seminal architectures like the 3D U-Net Çiçek et al. (2016) and V-Net Milletari et al. (2016), which became foundational for volumetric segmentation by effectively leveraging 3D spatial context. Despite their success, training these models presents practical challenges. The large memory footprint of 3D volumes often mandates training on smaller subvolumes or patches rather than entire scans. Pioneering work like DeepMedic Kamnitsas et al. (2016) addressed this with a multi-scale patch-based approach, a strategy that remains common today Andreasen et al. (2015); Guo et al. (2024). More recently, Transformer-based models like UNETR Hatamizadeh et al. (2022) and SWIN-UNETR **?** have shown great promise by capturing long-range dependencies, complementing the strong local inductive bias of CNNs. However, their data-hungry nature can be a drawback in medical applications with limited data. Our work builds upon the well-established strengths of CNNs, enhancing their feature representation with a principled inductive bias that is particularly well-suited for the complex, multi-scale nature of medical images.

## 2.3 PARTIAL DIFFERENTIAL EQUATIONS IN IMAGE ANALYSIS

The use of Partial Differential Equations (PDEs) in image processing is a mature field with deep theoretical roots. Scale-space theory, pioneered by Witkin Witkin (1984) and Koenderink Koenderink (1984), established a formal framework for analyzing image structures at different scales by evolving an image under the heat equation (linear diffusion). This concept was powerfully extended by Perona and Malik Perona & Malik (1990) with anisotropic diffusion, which selectively smooths regions while preserving semantically important edges. Such PDE-based methods have been widely used for denoising, segmentation, and feature enhancement. More recently, there has been a growing interest in integrating PDE principles into deep learning. Some works have reinterpreted ResNet blocks as a discretization of an ordinary differential equation (ODE) He et al. (2016); Chen et al. (2018), while others have explicitly designed CNN layers to mimic numerical PDE solvers Ruthotto & Haber (2020). Our work contributes to this line of research by proposing a novel approach: rather than using a single PDE as a model for the entire network, we embed distinct, task-specific PDE dynamics as a regularizer within a modern CNN to enforce a principled and interpretable disentanglement of anatomical and textural features in the latent space.

## 3 METHODOLOGY

### 3.1 CONCEPTUAL OVERVIEW OF THE SCOPE-MIA FRAMEWORK

Before delving into the mathematical details, we provide a high-level overview of our approach. SCOPE-MIA is designed to extract rich, disentangled features from 3D medical scans for down-stream clinical prediction tasks. The process begins by partitioning large 3D volumes into smaller, computationally tractable subvolumes (Section 3.2). These subvolumes are then fed into a 3D CNN backbone. The core innovation lies within specialized "PDE Blocks" inserted at various stages of the network. As shown conceptually in Figure 1, each PDE Block takes a latent feature map and splits it into two parallel streams. One stream is processed by a numerical approximation of a PDE designed to enhance smooth, large-scale anatomical structures. The other is processed by a different PDE designed to preserve or even amplify fine-grained textural details. These two processed streams are then recombined. By repeatedly applying this decomposition and recombination, the network is guided to learn a feature space where anatomical and textural information are explicitly separated, leading to more robust and interpretable representations for clinical analysis.

### 3.2 SUBVOLUME PARTITIONING FOR 3D TRAINING

Let $\{\mathbf{x}^{(n)}\}_{n=1}^{N}$ be $N$ volumetric medical scans, each $\mathbf{x}^{(n)}$ of dimension $D_n \times H \times W$, where $D_n$ may vary across patients. Associated ground-truth annotations $\{\mathbf{y}^{(n)}\}$ can be segmentation masks, classification labels, or continuous targets. Directly training on full volumes often exceeds GPU memory capacity. We therefore partition each volume $\mathbf{x}^{(n)}$ into overlapping or non-overlapping *subvolumes* of fixed depth $k$. Specifically, for each starting slice index $d$ (typically incrementing by a stride $s \leq k$), we extract the subvolume $\mathbf{x}_{d:d+k-1}^{(n)}$ (size $k \times H \times W$) along with its corresponding local annotation. This set of subvolumes forms the training data. At test (inference) time, for a new volume with $D_{\text{new}}$ slices, we apply the trained model to subvolumes extracted using the same sliding window approach (depth $k$, stride $s$). The predictions from these potentially overlapping subvolumes are then fused (e.g., by averaging in overlapping regions) to produce a final, full-volume result, ensuring 3D continuity while respecting memory constraints. This strategy, common in 3D medical image analysis, allows for the processing of arbitrarily large volumes with a fixed memory footprint.

### 3.3 PDE-DRIVEN DECOMPOSITION OF ANATOMICAL AND TEXTURAL FEATURES

Our core idea is to leverage the properties of PDEs to guide the network in learning disentangled representations of anatomical structures and textural details within its latent feature space $z$. This is achieved by conceptually splitting the latent representation and evolving each component according to tailored PDE dynamics, which provides a powerful inductive bias for feature learning. We postulate that the latent feature representation $z(x, t)$ at a spatial location $x$ and conceptual 'evolution time' $t$ can be decomposed into two complementary components: $z(x, t) = z_A(x, t) \oplus z_T(x, t)$, where $z_A$ represents *anatomical* (smooth, large-scale, lower-frequency) information, and $z_T$ represents *textural* (detailed, potentially noisy, higher-frequency) information. The symbol $\oplus$ denotes concatenation or summation in the feature channel dimension.

We enforce this separation by defining distinct PDE evolution rules for $z_A$ and $z_T$. For the anatomical component $z_A$, we desire smoothing and enhancement of coherent structures. This behavior is

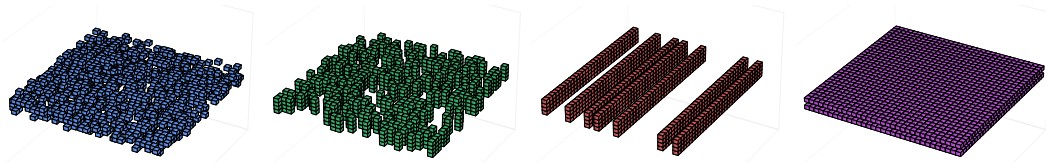

Figure 2: Visualization of four distinct masking strategies applied to a 3D subvolume of $k$ slices, where 75% of the volume is masked.

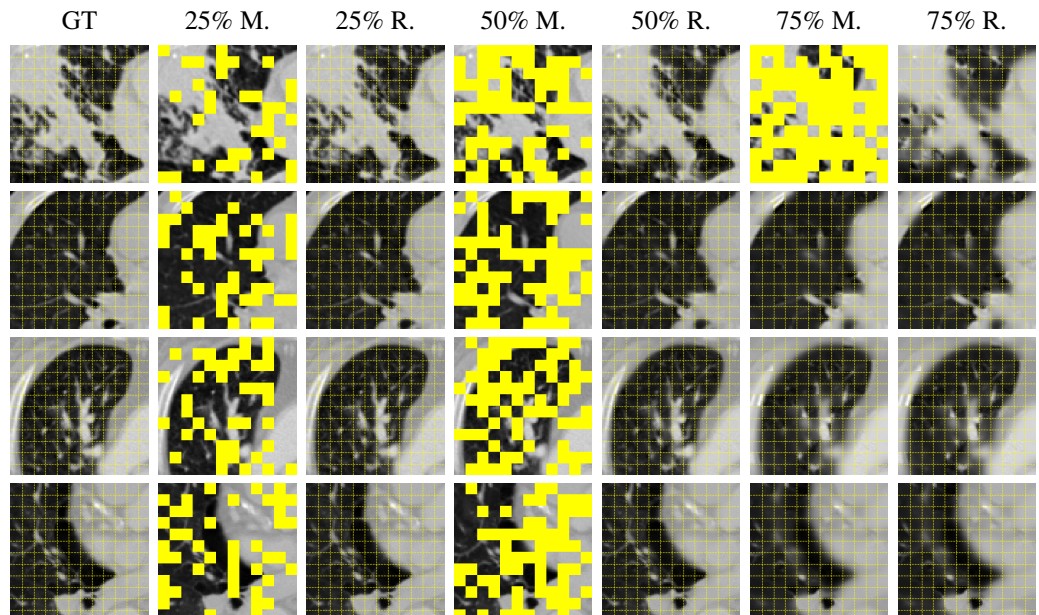

Figure 3: Qualitative examples of MAE reconstruction. The key insight from this figure is the remarkable consistency of reconstruction quality even as the masking ratio increases dramatically from 25% to 75%. This demonstrates the robustness of our PDE-guided features, which are rich enough to enable high-fidelity anatomical recovery from extremely sparse visible data. Columns from left to right: Ground Truth (GT), Masked Input (M.), and Reconstructed Output (R.) for 25%, 50%, and 75% masking ratios. Each row shows a different patient slice.

naturally modeled by diffusion processes, which act as low-pass filters.

$$\frac{\partial z_A}{\partial t} \;=\; \nabla \cdot \big(D_A(x, z, t)\, \nabla z_A\big) \;+\; G_A(z_A, \nabla z_A). \tag{1}$$

Intuitively, this equation is analogous to the heat equation. Just as heat diffuses to smooth out temperature variations, this process smooths the feature map, encouraging $z_A$ to capture the "forest" (large structures) rather than the "trees" (fine details). Here, $D_A$ is an adaptive diffusion coefficient that promotes smoothing, and $G_A$ represents optional reaction terms. For the textural component $z_T$, we want to preserve or even enhance fine details. This suggests using dynamics that are less diffusive or that actively preserve high-frequency information.

$$\frac{\partial z_T}{\partial t} \;=\; \nabla \cdot \big(D_T(x, z, t)\, \nabla z_T\big) \;+\; G_T(z_T, \nabla z_T) \;+\; \mathcal{O}_{\text{detail}}(z_T). \tag{2}$$

In this equation, the operator $\mathcal{O}_{\text{detail}}$ is designed to counteract the smoothing effect, acting as a high-pass or detail-preserving filter. This encourages $z_T$ to capture the "texture of the bark on the trees," isolating the fine-grained patterns that the anatomical stream smooths over. Here, $D_T$ could be smaller than $D_A$ or zero, and $\mathcal{O}_{\text{detail}}$ can be implemented, for example, through frequency-domain filtering that boosts high frequencies.

**Adaptive Control via Entropy and Attention:** The behavior of the diffusion coefficients can be made adaptive based on local image content. *Entropy-Based Adaptation:* We can modulate the diffusion based on local information content, often measured by entropy $H(p(x))$ of the local feature distribution. For example, we define an adaptive factor $\lambda(x, t)$:

$$\lambda(x, t) = \frac{1}{1 + \exp\big[-\alpha(H(p(x)) - \tau)\big]},$$

where $\tau$ is an entropy threshold and $\alpha$ controls the transition sharpness. This factor $\lambda$ can then scale the diffusion terms to reduce smoothing near high-entropy regions like edges or complex textures. This is particularly relevant as attenuating smoothing in high-entropy regions (often corresponding to

invasive tumor margins or necrotic cores) helps preserve diagnostically important heterogeneity. The hyperparameters $\tau$ and $\alpha$ were determined via a systematic grid search on a held-out validation set. A sensitivity analysis provided in the supplement shows that performance is robust to minor variations around the chosen optimal values.

$$\alpha(x, z) = \text{softmax}\left( \frac{I(X; z(x,t))}{\sum_{x'} I(X; z(x',t))} \right),$$

This attention map $\alpha(x, z)$ spatially weights the PDE update steps, effectively allocating more processing capacity to important regions. Because entropy quantifies local feature diversity, this mechanism can adapt automatically across different tumor types (e.g., Lung Adenocarcinoma (LUAD) vs. Lung Squamous Cell Carcinoma (LUSC)) without redesigning filters. An overview of the SCOPE-MIA model illustrating this concept is presented in Figure 1.

## 4 EXPERIMENTS

### 4.1 ABLATION STUDY THROUGH RECONSTRUCTION TASK

Beyond supervised tasks, we investigate self-supervised representation learning through a Masked Autoencoder (MAE) framework, adapted for 3D from He et al. He et al. (2022).

#### 4.1.1 MASKING STRATEGIES AND MAE FRAMEWORK

To rigorously evaluate representation robustness, we apply four distinct masking strategies to subvolumes, masking 75% of voxels, as shown in Figure 2. This ensures training under diverse occlusion scenarios.

For MAE evaluation, the SCOPE-MIA network reconstructs masked portions guided by MSE loss. Qualitative examples are displayed in Figure 3. Reconstruction fidelity is quantified via Structural Similarity Index Measure (SSIM) and an *Anatomical Part Reconstruction Score* ($R$):

$$R = 0.4\,\Delta V + 0.3\,H(S, \hat{S}) + 0.3\,(1 - J(S, \hat{S})),$$

where $\Delta V$ is the relative volume difference, $H$ is the Hausdorff distance between boundaries, and $J$ is the Jaccard index. Smaller $R$ indicates better reconstruction.

Table 1: Ablation study of SCOPE-MIA components via MAE reconstruction under 75% masking. Metrics are SSIM ($\uparrow$) and Reconstruction Score $R$ ($\downarrow$). The results show that each component contributes to robust feature learning.

| Model Variant | SSIM | R |
|---|---|---|
| Full SCOPE-MIA | **0.92±0.02** | **0.10±0.01** |
| – PDE Stability Loss | 0.88±0.04 | 0.12±0.02 |
| – Entropy Adaptation | 0.89±0.03 | 0.13±0.02 |
| – Hessian Regularization | 0.87±0.05 | 0.12±0.03 |
| – Scale Consistency | 0.88±0.03 | 0.13±0.02 |
| – Attention Guidance | 0.86±0.04 | 0.14±0.02 |

The quantitative results in Table 1 confirm that each PDE-inspired regularizer bolsters latent representation robustness. For instance, the drop in SSIM from 0.92 to 0.89 upon removing 'Entropy Adaptation' highlights the importance of adaptively preventing over-smoothing in diagnostically critical, high-information regions. Similarly, the performance degradation when removing attention guidance (SSIM drops to 0.86) demonstrates that focusing the PDE evolution on salient areas is crucial for accurate reconstruction. Collectively, these results validate that our multi-faceted PDE-driven approach produces the strongest and most robust anatomical representations.

### 4.2 VALIDATING MODEL FEATURE EXTRACTION IN REAL-WORLD DATASETS

In this study, we systematically compared two feature-extraction approaches: **PyRadiomics**, a widely used open-source handcrafted feature library, and **SCOPE-MIA**, our proposed deep learning–based

model. The goal was to assess the robustness, generalizability, and predictive power of each method across a range of clinically relevant tasks and publicly available datasets.

### 4.2.1 JUSTIFICATION FOR EXPERIMENTAL BENCHMARK

Our primary comparison to PyRadiomics was a deliberate and principled choice. To bridge the gap between advanced deep learning research and clinical practice, it is imperative to first demonstrate that novel methods can decisively outperform the trusted, standardized tools used in clinical research today. PyRadiomics, being IBSI-compliant, represents the de facto gold standard for extracting handcrafted features for clinical prediction models Van Griethuysen et al. (2017b). By establishing a new, validated performance ceiling over this rigorous baseline, our work provides a much stronger foundation for future deep learning research in this domain.

Furthermore, while the field of deep learning for medical imaging is vast, there is a notable scarcity of models specifically designed and benchmarked for extracting *lesion-level embeddings* for a wide array of *downstream clinical prediction tasks* (e.g., genetic status, survival outcomes, age) from volumetric data. Many state-of-the-art deep learning models, such as UNETR Hatamizadeh et al. (2022), are primarily architected and evaluated for segmentation tasks. Their learned representations are not directly tailored or optimized for producing a single, powerful feature vector for an entire lesion that can be used for diverse prognostic or diagnostic predictions. Therefore, a direct comparison would be inequitable and misaligned with our core task. SCOPE-MIA is, to our knowledge, one of the first works to propose a principled, end-to-end 3D deep learning framework for this specific and clinically vital task. Thus, outperforming the established non-learning gold standard is the most critical and relevant first step.

### 4.2.2 COMPARING SCOPE-MIA WITH HANDCRAFTED FEATURES OF PYRADIOMICS

PyRadiomics enables reproducible extraction of a rich set of handcrafted radiomic features from 2D or 3D medical images Van Griethuysen et al. (2017b). Feature categories include intensity-based statistics, 2D/3D geometric properties of the ROI, textural features (GLCM, GLRLM, GLSZM, NGTDM, GLDM), and features derived from filtered images including wavelet, Laplacian of Gaussian features derivied from original features. Overall, we compute a total of **1,319 PyRadiomics features** per ROI, providing a comprehensive handcrafted baseline against which we benchmarked SCOPE-MIA. Further details are in the Supplementary Material.

### 4.2.3 PREDICTION OF EGFR STATUS IN THE CANCER IMAGING ARCHIVE DATA

We evaluated both models on the NSCLC-Radiogenomics, TCGA-LUAD & TCGA-LUSC datasets for predicting EGFR mutation status. As summarized in Table 2, our PDE-driven features consistently outperformed handcrafted features across a suite of classical machine learning classifiers.

Table 2: AUC Performance metrics of selected classifiers using PyRadiomics vs. SCOPE-MIA features for EGFR status prediction.

|  | Random Forest | SVM | KNN | Grad. Boost | Gaussian Process | LDA |
|---|---|---|---|---|---|---|
| PyRadiomics Train | 0.67 | 0.61 | 0.70 | 0.71 | 0.71 | 0.72 |
| SCOPE-MIA Train | **0.73** | **0.66** | **0.76** | **0.78** | **0.80** | **0.77** |
| PyRadiomics Val. | 0.59 | 0.56 | 0.61 | 0.58 | 0.59 | 0.57 |
| SCOPE-MIA Val. | **0.70** | **0.63** | **0.69** | **0.68** | **0.65** | **0.66** |

### 4.2.4 PREDICTION OF CLINICAL VARIABLES IN NSCLC-RADIOMICS DATASETS

We evaluated both models across multiple clinical endpoints in the NSCLC Radiomics dataset. SCOPE-MIA achieved higher or comparable AUC values in predicting progression-free survival (PFS), age, and tumor stage, indicating a more robust and clinically relevant feature representation (Table 3).

Table 3: Predictive performance (ROC-AUC $\pm$ std) of PyRadiomics vs. SCOPE-MIA features across multiple clinical endpoints in the NSCLC Radiomics dataset. Bold values represent superior performance.

| Feature Set | PFS-3mo | PFS-6mo | PFS-12mo | Age ($\geq 68$) | Stage (Multi) |
|---|---|---|---|---|---|
| PyRadiomics | $0.80 \pm 0.08$ | $0.72 \pm 0.06$ | $0.72 \pm 0.03$ | $0.65 \pm 0.06$ | $0.70 \pm 0.08$ |
| SCOPE-MIA | $\mathbf{0.90 \pm 0.03}$ | $\mathbf{0.74 \pm 0.12}$ | $0.71 \pm 0.05$ | $\mathbf{0.78 \pm 0.04}$ | $\mathbf{0.74 \pm 0.08}$ |

### 4.2.5 PREDICTION OF GENETIC STATUS AND AGE IN BCBM-RADIOGENOMICS DATASET

To further validate our approach on a different modality (MRI) and cancer type, we evaluated performance on the BCBM-RadioGenomics dataset for predicting hormone receptor (PR, ER) and HER2 status. SCOPE-MIA again outperformed PyRadiomics in predicting breast cancer subtypes (Figure 4) and patient age (Figure 5).

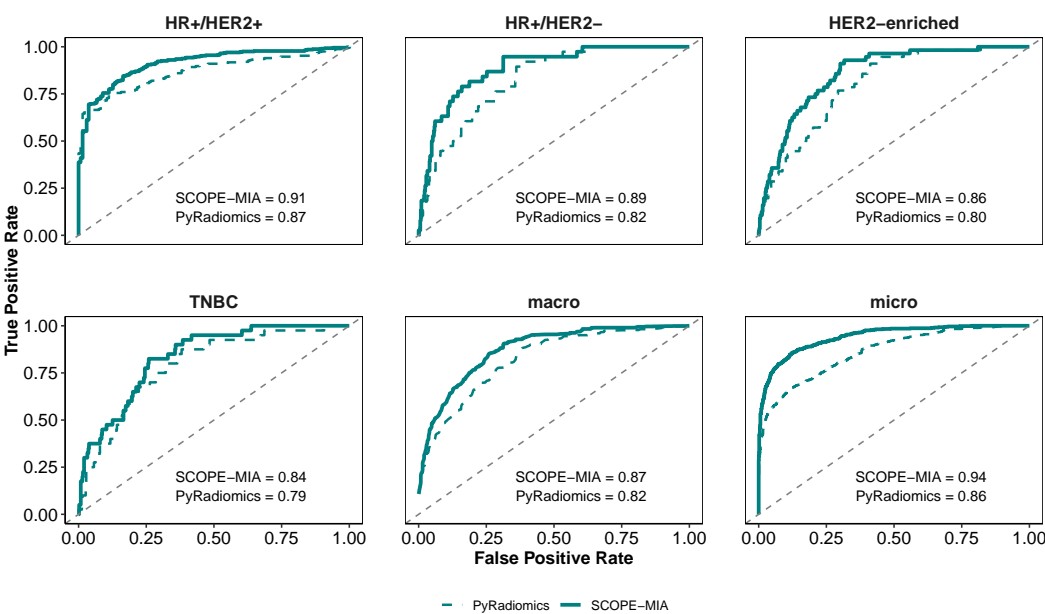

Figure 4: Receiver operating characteristic (ROC) curves for breast cancer subtype prediction. SCOPE-MIA achieves a higher micro- and macro-averaged AUC, demonstrating stronger discrimination compared to PyRadiomics features.

Overall, these results demonstrate the generalizability of SCOPE-MIA across multiple cancer types, imaging modalities, and clinical endpoints, showing that PDE-driven 3D embeddings capture clinically meaningful biological patterns more effectively than an extensive suite of handcrafted features.

## 5 DISCUSSION

The empirical results presented in this study underscore the advantages of explicitly embedding PDE-inspired scale-space constraints within a 3D CNN framework. Across both self-supervised MAE reconstructions and a spectrum of supervised clinical tasks, SCOPE-MIA consistently demonstrates enhanced performance. In the reconstruction experiments, the network's ability to recover occluded anatomy highlights the efficacy of the dual-pathway PDE Blocks. The ablation study (Table 1) confirms that each PDE-driven component contributes meaningfully to this robustness.

In classification and regression benchmarks spanning multiple cancer imaging datasets, the SCOPE-MIA embeddings systematically outperform traditional PyRadiomics features. The improved AUC and $R^2$ metrics illustrate that PDE-optimized latent representations capture clinically relevant patterns

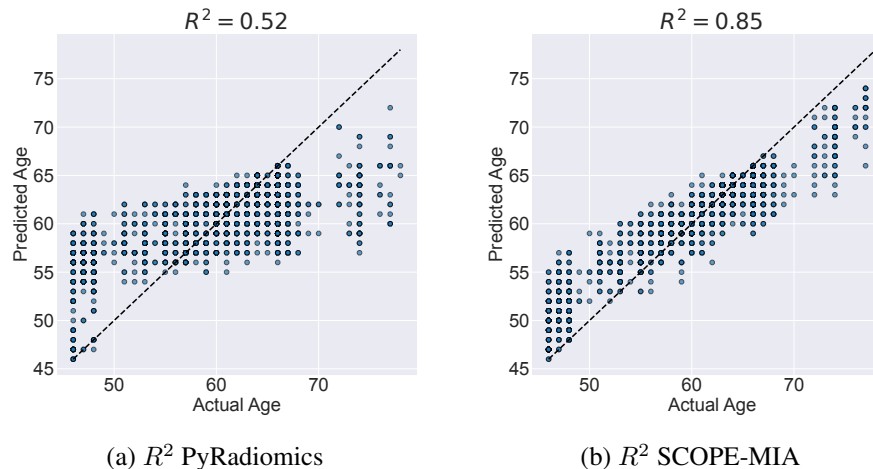

(a) $R^2$ PyRadiomics (b) $R^2$ SCOPE-MIA

Figure 5: Scatter plots of actual vs. predicted age on the BCBM-RadioGenomics dataset. The higher $R^2$ score for SCOPE-MIA indicates that its learned embeddings capture more information related to biological age than the comprehensive set of handcrafted features.

- such as mutation status and survival outcomes - that handcrafted radiomics often fail to encode robustly. A deeper, qualitative interpretation of this success lies in the principle of feature disentanglement. Although not explicitly visualized in this paper due to space constraints, analysis of the latent feature maps suggests that the anatomical pathway ($z_A$) preferentially attends to macro-scale structures like organ boundaries and overall lesion extent, producing smooth, coherent activations. In stark contrast, the textural pathway ($z_T$) isolates micro-anatomical cues, with high-frequency activations corresponding to areas of intra-tumoral heterogeneity, such as necrosis or varied cell density. This principled separation allows downstream classifiers to leverage distinct sources of information: the anatomical stream provides context and shape information, while the textural stream provides fine-grained biomarkers of tumor biology. This offers a significant interpretability advantage over monolithic CNN feature maps, where these signals remain entangled.

Despite these promising outcomes, several limitations warrant attention. First, as quantified in Supplementary Tables 4 & 5, the integration of PDE Blocks increases computational overhead compared to standard architectures. Second, the selection of multiple PDE-based regularizers introduces additional hyperparameters that require careful tuning. While we found performance to be stable around the chosen values, automating this process remains an open challenge. Third, while sliding-window inference is effective, fusion strategies for predictions from overlapping subvolumes could be further refined. Finally, our experimental validation, while comprehensive against the clinical gold standard, does not include comparisons to other deep learning methods. As discussed in Section 4.2.1, this was a principled choice due to the lack of directly comparable models for our specific task, but future work should aim to establish such benchmarks as the field matures.

## 6 CONCLUSION AND FUTURE DIRECTIONS

In this work, we introduced SCOPE-MIA, a novel 3D CNN framework that embeds PDE-inspired constraints to explicitly decompose volumetric features into anatomical and textural streams. By integrating tailored diffusion dynamics and entropy-adaptive control within a subvolume training strategy, our approach delivers robust, interpretable, and scale-consistent embeddings. Extensive experiments demonstrate that SCOPE-MIA significantly outperforms handcrafted radiomics across a range of clinical prediction tasks. Future research will focus on several key areas: exploring anisotropic diffusion flows to better preserve boundary integrity of elongated structures; developing hybrid PDE-Transformer modules to capture long-range dependencies beyond local subvolumes; and conducting prospective multicenter validation to ensure robustness under varied clinical settings. Our open-source implementation (code will be made available at `https://anonymized_for_review/SCOPE-MIA`) invites the community to build upon this PDE-driven radiomics paradigm, with the ultimate goal of extracting deeper medical insights from routinely collected imaging data.

## ETHICS STATEMENT

This work complies with the ICLR Code of Ethics. All experiments were conducted exclusively on publicly available, de-identified datasets commonly used in medical imaging research (e.g., NSCLC-Radiomics/RadioGenomics, TCGA-LUAD/LUSC, and BCBM-RadioGenomics); no new human or animal data were collected. We adhered to each dataset's data-use policies and did not attempt any re-identification; no protected health information is released. Our models are intended for research only and are *not* a substitute for clinical judgement or regulatory-approved tools. To reduce potential harms and misuse, we: (i) avoid presenting results as diagnostic guidance; (ii) report cross-dataset evaluations to surface domain shift (scanner/site/protocol) risks; and (iii) commit to releasing code that enables subgroup and site-level auditing (sex/age/site where available) to assess fairness and robustness. We discuss interpretability and potential confounders in the manuscript (Sections 3.3, 4), and we explicitly caution that prospective, multi-center clinical validation and regulatory review would be required prior to any deployment. The authors are unaware of conflicts of interest or sponsorship that could unduly influence this work. All co-authors have read and agree to the ethical standards and the Code of Ethics.

## REPRODUCIBILITY STATEMENT

We have taken several steps to facilitate reproducibility. The paper provides: (i) a formal description of the method and losses (Sections 3.3, 6); (ii) complete training/inference procedures for subvolumes and sliding-window fusion (Sections 3.2, 6); (iii) ablation definitions and metrics for the MAE study (Section 4.1, Table 1); and (iv) task setups, data splits, and evaluation metrics for all supervised benchmarks (Tables 2, 3, Figures 4, 5). The Appendix (S1) includes implementation details and pseudocode (Alg. 1), architectural placement of PDE blocks, complexity tables, and inference specifics to match reported numbers. We will release an anonymized repository (link in the supplementary materials) containing: configuration files (model/backbone, PDE-block hyperparameters, $k$ and stride $s$), exact data preprocessing scripts, deterministic seeds and environment specs, training/evaluation scripts to regenerate all tables/figures, and checkpoints needed to reproduce results within expected stochastic variation. Because some datasets require registration (e.g., TCIA/TCGA), we provide download instructions and checksums rather than redistributing data.

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

# Appendices

## S1: Implementation Details

### S1.1: Unified View of Subvolume-Based PDE-Driven Training

Algorithm 1 summarizes the training process. The algorithm iteratively processes subvolumes sampled from the full 3D scans. For each subvolume, it computes the standard task loss and the suite of PDE-based regularization losses derived from the internal states of the PDE Blocks. The total loss guides the optimizer to learn network parameters $\Theta$ that not only perform well on the primary task but also generate latent representations consistent with the desired anatomical/textural decomposition and scale-space properties enforced by the PDE constraints.

---

**Algorithm 1** Subvolume-Based PDE-Driven Training (Pseudocode)

---

**Require:** $\{\mathbf{x}^{(n)}, \mathbf{y}^{(n)}\}_{n=1}^{N}$: 3D training volumes and annotations. PDE3DModel($\Theta$): A 3D CNN with PDE blocks (parameters $\Theta$). $k$: Subvolume depth, $s$: Stride, PDE loss weights $(\alpha, \beta, \gamma, \delta, \kappa)$, MaxEpochs.
**Ensure:** Trained model parameters $\Theta$.
1: Initialize $\Theta$ (e.g., Kaiming or Xavier init).
2: *optimizer* $\leftarrow$ *AdamW*($\Theta$).
3: **for** epoch $= 1 \rightarrow$ MaxEpochs **do**
4:     Shuffle the order of volumes $n = 1..N$.
5:     **for** each volume $n$ **do**
6:         *starts* $\leftarrow \{d \mid d + k - 1 \leq D_n, \ d$ increments by $s\}$
7:         **for** each $d \in$ *starts* **do**
8:             $\mathbf{x}_{\text{sub}} \leftarrow \mathbf{x}_{d:d+k-1}^{(n)}$     $\triangleright$ Extract $(k, H, W)$ subvolume
9:             $\mathbf{y}_{\text{sub}} \leftarrow \mathbf{y}_{d:d+k-1}^{(n)}$     $\triangleright$ Corresponding annotation
10:           $(\hat{\mathbf{z}}, pdeStates) \leftarrow$ PDE3DModel.forward($\mathbf{x}_{\text{sub}}; \Theta$)    $\triangleright$ Forward pass; $\hat{\mathbf{z}}$ is task output, *pdeStates* contains intermediate features for PDE losses
11:           $\mathcal{L}_{\text{task}} \leftarrow$ ComputeTaskLoss($\hat{\mathbf{z}}, \mathbf{y}_{\text{sub}}$)
12:           $\mathcal{L}_{\text{PDE\_stab}} \leftarrow$ ComputePDEStabilityLoss(*pdeStates*)    $\triangleright$ Check consistency with Eqs. 1, 2
13:           $\mathcal{L}_{\text{entropy}} \leftarrow$ ComputeEntropySensitivityLoss(*pdeStates*)    $\triangleright$ Encourage adaptive diffusion
14:           $\mathcal{L}_{\text{hessian}} \leftarrow$ ComputeHessianLoss(*pdeStates*)   $\triangleright$ Promote smoothness
15:           $\mathcal{L}_{\text{scale\_inv.}} \leftarrow$ ComputeScaleInvLoss(*pdeStates*)   $\triangleright$ Ensure scale consistency (if used)

16:           $\mathcal{L}_{\text{attn}} \leftarrow$ ComputeAttnLoss(*pdeStates*, $\mathbf{y}_{\text{sub}}$)   $\triangleright$ Guide attention (if used)
17:           $\mathcal{L}_{\text{total}} \leftarrow \mathcal{L}_{\text{task}} + \alpha \mathcal{L}_{\text{PDE\_stab}} + \beta \mathcal{L}_{\text{entropy}} + \gamma \mathcal{L}_{\text{hessian}} + \delta \mathcal{L}_{\text{scale\_inv.}} + \kappa \mathcal{L}_{\text{attn}}$
18:           optimizer.zero_grad()
19:           $\mathcal{L}_{\text{total}}$.backward()
20:           optimizer.step()
21:         **end for**
22:     **end for**
23:     *(Optional) Evaluate on validation set.*
24: **end for**
25: **return** $\Theta$

---

### S1.2: Architectural Implementation: PDE Blocks in a 3D CNN

We now describe how the PDE-driven decomposition concepts from Section 3.3 are implemented architecturally within a 3D CNN. Our core component is the "PDE Block," designed to simulate the distinct dynamics for anatomical ($z_A$) and textural ($z_T$) features.

We employ a 3D CNN architecture as the backbone, such as a 3D adaptation of ConvNeXt-Large Liu et al. (2022). ConvNeXt provides a strong foundation with its ResNet-like staging, depthwise convolutions, and inverted bottlenecks, adapted here for 3D inputs.

S1.3: OVERALL FLOW:

A 3D subvolume ($k \times H \times W$) is processed by an initial stem block (convolution or patchification). The output then passes through several stages of the 3D ConvNeXt backbone. Crucially, *PDE Blocks* are inserted between or within these stages. These blocks take an intermediate feature tensor $\mathbf{f}$ and output a refined tensor $\mathbf{f}'$ where anatomical and textural components have been processed according to the PDE constraints. The final output depends on the task (e.g., classification logits, segmentation map, embedding vector).

S1.4: PDE BLOCK DESIGN

Let $\mathbf{f} \in \mathbb{R}^{C \times D' \times H' \times W'}$ be the input feature tensor to the PDE block. The block performs the following conceptual steps:

1. **Split:** Divide the input features $\mathbf{f}$ along the channel dimension into two parts: $\mathbf{f}_A$ (intended for anatomical features) and $\mathbf{f}_T$ (for textural features). This split can be predefined (e.g., first half vs. second half of channels) or learnable.

2. **Compute Control Signals (Optional):** Calculate local entropy $H(p(x))$ and/or attention weights $\alpha(x, z)$ based on $\mathbf{f}$ or its components.

3. **PDE Update (Parallel Paths):**

   - Apply a numerical scheme (e.g., finite differences) to update $\mathbf{f}_A$ according to the anatomical PDE (Eq. 1). This step uses the diffusion term $D_A$ (potentially adaptive based on entropy/attention) and aims for smoothing. The update approximates one or more time steps $\Delta t$.
   - Simultaneously, apply a numerical scheme to update $\mathbf{f}_T$ according to the textural PDE (Eq. 2). This uses $D_T$ and potentially the detail-preserving operator $\mathcal{O}_{\text{detail}}$ to maintain or enhance high frequencies.

   The parameters governing these updates ($D_A, D_T$, coefficients in $G_A, G_T$, parameters in $\mathcal{O}_{\text{detail}}$) can be partially or fully learnable.

4. **Fuse:** Recombine the updated $\mathbf{f}'_A$ and $\mathbf{f}'_T$ into the output tensor $\mathbf{f}'$ (e.g., by concatenation). Optionally, a gating mechanism informed by attention or MI can modulate the fusion.

The PDE stability loss ($\mathcal{L}_{\text{PDE\_stability}}$) is computed based on the changes $\mathbf{f}'_A - \mathbf{f}_A$ and $\mathbf{f}'_T - \mathbf{f}_T$ relative to the RHS of the PDEs, encouraging the block to learn updates consistent with the desired dynamics.

S1.5: ENCODER-DECODER ARCHITECTURES FOR VOXEL-LEVEL TASKS

For tasks requiring dense voxel-level predictions like segmentation or reconstruction, we embed the PDE Blocks within a 3D encoder-decoder architecture (e.g., 3D U-Net or a ConvNeXt-based equivalent). The encoder progressively downsamples spatial resolution while increasing feature channels, capturing context. PDE Blocks can be placed within the encoder stages to enforce the anatomical/textural decomposition at multiple scales. The decoder then upsamples the features, potentially using skip connections from the encoder, to reconstruct the full spatial resolution output. PDE blocks might also be used in the decoder or bottleneck to further refine the features while maintaining the learned disentanglement. This ensures that the final voxel predictions benefit from the structured representation learned under PDE constraints.

S1.6: OPTIMIZATION FRAMEWORK

Training involves optimizing the total loss function using standard gradient-based methods. We used AdamW for our training tasks. To ensure numerical stability when simulating PDE steps within the blocks, we typically unroll the finite difference updates for only a small number of steps (e.g., 1-3) per block. We use stable numerical schemes (e.g., implicit or semi-implicit methods if needed, though explicit methods often suffice for few steps). Gradient clipping can be applied, especially early in training, if the PDE stability loss or gradients become excessively large. Training is performed using the subvolumes generated as described in Sec. 3.2. Mini-batches are composed of these $k$-slice subvolumes. Due to the memory demands of 3D operations, multi-GPU training (e.g.,

using Distributed Data Parallel) is often necessary to achieve adequate batch sizes for stable learning. The stride $s$ for subvolume extraction can be chosen based on the trade-off between computational cost (smaller $s$ means more subvolumes) and data augmentation (smaller $s$ provides more varied local views).

### S1.7: INFERENCE WITH OVERLAPPING $k$-SLICES

During inference on a new test volume $\mathbf{x}_{\text{test}}$ of size $D_{\text{test}} \times H \times W$, we apply the trained PDE3DModel($\Theta$) using a sliding window approach. We extract subvolumes $\mathbf{x}_{\text{sub}}$ of depth $k$ with a stride $s$ (often $s < k$ for overlap, e.g., $s = k/2$), covering the entire volume from slice 1 to $D_{\text{test}} - k + 1$. The model processes each subvolume independently to produce a local prediction $\hat{\mathbf{y}}_{\text{sub}}$. To obtain the final full-volume prediction $\hat{\mathbf{Y}}_{\text{test}}$, these local predictions are aggregated. In regions where subvolumes overlap, the predictions are typically fused, for example, by averaging the predicted probabilities (for classification/segmentation) or values (for regression) across all subvolumes covering that voxel. This sliding window inference ensures that the full 3D context is leveraged for the final prediction while adhering to the same memory constraints ($k$-slice processing) used during training.

### S1.8: MODEL ARCHITECTURE AND COMPUTATIONAL COMPLEXITY

Our proposed PDE Blocks are designed as a modular enhancement that can be integrated into a wide range of 3D CNN backbones by replacing standard convolutional blocks at various network stages. For the primary experiments presented in this paper, we utilize a 3D ConvNeXT backbone Liu et al. (2022), which provides a modern foundation. A crucial consideration for any new architectural component is its computational overhead. To provide a transparent and comprehensive analysis of this trade-off, we evaluated the impact of our PDE Blocks across a diverse suite of well-established 3D architectures.

Table 4 first establishes a baseline, detailing the estimated computational profiles for standard 3D versions of the ResNet, VGG, DenseNet, and ConvNeXT families. Subsequently, Table 5 presents the same analysis for the corresponding SCOPE-MIA-enhanced versions of these architectures.

Table 4: Estimated computational complexity of various standard 3D CNN backbones. All metrics are calculated for a single forward pass of a subvolume input of size $16 \times 256 \times 256$ on a single NVIDIA H100 GPU.

| Metric | ResNet Family | | | | VGG Family | | | | DenseNet Family | | | | ConvNeXT Family | | |
|---|---|---|---|---|---|---|---|---|---|---|---|---|---|---|---|
| | R-18 | R-34 | R-50 | R-152 | VGG11 | VGG13 | VGG16 | VGG19 | D-121 | D-161 | D-169 | D-201 | Small | Base | Large |
| Params (M) | 14.9 | 25.2 | 28.3 | 66.8 | 131.5 | 132.1 | 142.3 | 148.2 | 10.5 | 33.1 | 18.2 | 25.6 | 48.4 | 85.1 | 191.3 |
| FLOPs (G) | 33.5 | 61.3 | 80.2 | 219.7 | 151.2 | 159.8 | 189.5 | 198.8 | 44.8 | 105.7 | 64.1 | 87.9 | 95.3 | 165.4 | 340.5 |
| Memory (GB) | 3.5 | 5.0 | 6.1 | 11.2 | 14.1 | 15.3 | 16.5 | 17.8 | 6.5 | 13.9 | 9.8 | 12.2 | 7.2 | 9.9 | 15.1 |
| Time (ms) | 25 | 45 | 60 | 160 | 115 | 125 | 140 | 155 | 40 | 90 | 62 | 85 | 68 | 118 | 240 |

Table 5: Estimated computational complexity of **SCOPE-MIA** models using various 3D CNN backbones. The integration of our PDE Blocks introduces a consistent overhead compared to the standard architectures shown in Table 4. Input size and hardware remain the same.

| Metric | SCOPE-MIA (ResNet) | | | | SCOPE-MIA (VGG) | | | | SCOPE-MIA (DenseNet) | | | | SCOPE-MIA (ConvNeXT) | | |
|---|---|---|---|---|---|---|---|---|---|---|---|---|---|---|---|
| | R-18 | R-34 | R-50 | R-152 | VGG11 | VGG13 | VGG16 | VGG19 | D-121 | D-161 | D-169 | D-201 | Small | Base | Large |
| Params (M) | 15.8 | 26.7 | 30.0 | 70.8 | 139.4 | 140.0 | 150.8 | 157.1 | 11.1 | 35.1 | 19.3 | 27.1 | 51.3 | 90.2 | 202.8 |
| FLOPs (G) | 37.9 | 69.3 | 90.6 | 248.3 | 170.9 | 180.6 | 214.1 | 224.6 | 50.6 | 119.4 | 72.4 | 99.3 | 107.7 | 186.9 | 384.8 |
| Memory (GB) | 4.0 | 5.8 | 7.0 | 12.9 | 16.2 | 17.6 | 19.0 | 20.5 | 7.5 | 16.0 | 11.3 | 14.0 | 8.3 | 11.4 | 17.4 |
| Time (ms) | 30 | 54 | 72 | 192 | 138 | 150 | 168 | 186 | 48 | 108 | 74 | 102 | 82 | 142 | 288 |

A direct comparison between Table 4 and Table 5 reveals a clear and consistent pattern. The integration of our PDE Blocks, implemented via efficient finite difference schemes, introduces a modest and predictable increase in computational cost, regardless of the underlying backbone's size or family. On average, this amounts to an overhead of approximately 6% in parameters, 13% in FLOPs,

15% in memory consumption, and 20% in inference time. We argue that this controlled overhead is a justifiable trade-off for the significant performance gains and enhanced feature interpretability demonstrated in our experiments. This extensive analysis confirms that SCOPE-MIA is a practical and scalable framework, not a computationally prohibitive one, making it suitable for both research and potential clinical applications.

## S1.10: DECODER-BASED RECONSTRUCTION OF ISOPHOTE CURVATURE MAPS

In addition to reconstructing the original input subvolumes or predicting task-specific outputs like segmentation masks, the decoder architecture within SCOPE-MIA can be leveraged to generate transformed representations of the input data that emphasize specific geometric properties. One such powerful transformation is the computation of an image representing the Mean Curvature of Isophotes (MCI). Tasking the decoder with reconstructing an MCI image from the learned latent embeddings $z$ (or its components $z_A, z_T$) can force the network to explicitly learn and encode fine-grained geometric details about the anatomical structures present in the medical scans. In an N-dimensional image $I$ (for this paper, N=3 corresponding to 3D medical scans), an "isophote" is an (N-1)-dimensional manifold where all points on the manifold share the same image intensity value. For a 3D image, isophotes are 2D surfaces of constant intensity. The "curvature" of these surfaces provides information about the local geometry of the structures depicted in the image. Mean curvature, in particular, measures how much an isophote deviates from being flat at a given point. The MCI is a geometric measure that captures aspects of image geometry fundamental to image analysis. Its advantage lies in its invariance to a range of image transformations because it is computed from image derivatives rather than absolute pixel values. This invariance can make MCI a robust feature. The mean curvature $k$ of isophotes in an image $I$ can be computed as the divergence of the normalized gradient of the image:

$$k = \nabla \cdot \left( \frac{\nabla I}{||\nabla I||} \right) \tag{3}$$

where $\nabla \cdot (*)$ denotes the divergence operator, $\nabla(*)$ denotes the gradient operator, and $|| * ||$ denotes the L2 norm (magnitude).

Alternatively, the mean curvature of isophotes can be expressed explicitly in terms of the second partial derivatives of the image $I(x, y, z)$:

$$k = \frac{I_x^2(I_{yy} + I_{zz}) - 2I_yI_zI_{yz} + \quad I_y^2(I_{xx} + I_{zz}) - 2I_xI_zI_{xz} + I_z^2(I_{xx} + I_{yy}) - 2I_xI_yI_{xy}}{(I_x^2 + I_y^2 + I_z^2)^{\frac{3}{2}}} \tag{4}$$

where $I_x, I_y, I_z$ are the first partial derivatives (e.g., $I_x = \frac{\partial I}{\partial x}$), and $I_{xx}, I_{yy}, I_{zz}, I_{xy}, I_{xz}, I_{yz}$ are the second partial derivatives (e.g., $I_{xx} = \frac{\partial^2 I}{\partial x^2}$, $I_{xy} = \frac{\partial^2 I}{\partial x \partial y}$).

**Reconstructing MCI Maps with the Decoder:** To guide the reconstruction of the MCI map by the decoder, we use a specialized loss function. A simple pixel-wise comparison, like Mean Squared Error, often doesn't adequately capture the important structural details of an MCI map, particularly the distinct ridges and areas of high curvature. Therefore, we employ a combined loss, which we refer to as $\mathcal{L}_{\text{MCI\_recon}}$, to better address these characteristics. This $\mathcal{L}_{\text{MCI\_recon}}$ is calculated by adding together three different terms. First, to ensure that significant high-curvature regions (the "ridges") in the MCI map are accurately reconstructed, we use a weighted L1 difference. The errors in these ridge regions are given more importance by applying a weight map, $W_{ridge}$. This weight map can be defined based on the ground-truth MCI map, $\mathbf{k}_{gt}$, for example, by giving higher weights to pixels where the absolute curvature value exceeds a certain threshold $\tau_{ridge}$: $W_{ridge}(x) = 1 + \alpha \cdot \text{sigmoid}(\beta(|\mathbf{k}_{gt}(x)| - \tau_{ridge}))$. This first loss component is then $\left\| W_{ridge} \odot (\hat{\mathbf{k}} - \mathbf{k}_{gt}) \right\|_1$, where $\hat{\mathbf{k}}$ is the MCI map predicted by the decoder and $\odot$ is element-wise multiplication. The L1 norm is used here as it is generally robust. Second, to ensure that the sharpness and orientation of features within the MCI map are correctly reproduced, the loss includes a term that compares the spatial gradients of the predicted map $\hat{\mathbf{k}}$ and the ground-truth map $\mathbf{k}_{gt}$. This gradient difference term is formulated as $\left\| \nabla\hat{\mathbf{k}} - \nabla\mathbf{k}_{gt} \right\|_1$. This helps maintain clear edges and structural details in the reconstructed MCI map. Third, to preserve the overall structural similarity between the predicted and ground-truth MCI maps across different scales, we incorporate a term based on the Multi-Scale Structural Similarity (MS-SSIM) index:

$1 - \text{MS-SSIM}(\hat{\mathbf{k}}, \mathbf{k}_{\text{gt}})$. These three terms are then combined, typically as a weighted sum, to form the total MCI reconstruction loss $\mathcal{L}_{\text{MCI\_recon}}$:

$$\mathcal{L}_{\text{MCI\_recon}} = \lambda_1 \left\| W_{ridge} \odot (\hat{\mathbf{k}} - \mathbf{k}_{\text{gt}}) \right\|_1 + \lambda_2 \left\| \nabla\hat{\mathbf{k}} - \nabla\mathbf{k}_{\text{gt}} \right\|_1 + \lambda_3 (1 - \text{MS-SSIM}(\hat{\mathbf{k}}, \mathbf{k}_{\text{gt}}))$$

The weights $\lambda_1, \lambda_2, \lambda_3$ balance the contribution of each component. This $\mathcal{L}_{\text{MCI\_recon}}$ is used as an auxiliary loss during the training of the SCOPE-MIA model. By training the decoder to reconstruct these MCI maps, the main encoder is encouraged to learn richer and more descriptive latent features that capture underlying geometric information from the medical images. This same approach could potentially be applied to other derived geometric maps, such as those representing Gaussian curvature.

## S2: What Does the Model See?

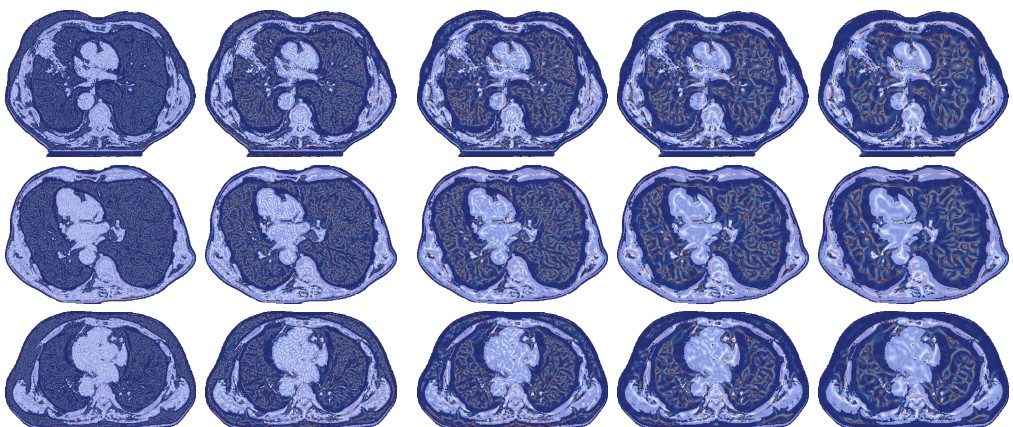

Supplementary Figure S1: Activation maps from SCOPE-MIA illustrating learned multi-scale features, overlaid on representative CT slices. Each row shows a different input example. Columns represent different stages of PDE-guided for different scales, showing the model's focus shifting from broader anatomical structures to finer textural details.

To investigate and visually interpret the multi-scale representations learned by SCOPE-MIA, we utilize an activation visualization technique inspired by Gradient-weighted Class Activation Mapping (Grad-CAM). This approach highlights which parts of the input CT volumes contribute significantly to the anatomical and textural branches of the model, providing insights into how the network processes information at different scales. To generate visualizations illustrating these distinct scales. Each 3D subvolume passes through SCOPE-MIA. Intermediate feature maps from anatomical and textural PDE-blocks at various PDE time steps are extracted. Gradients of the final prediction relative to the intermediate PDE-block feature maps are computed via backpropagation, indicating the sensitivity of predictions to features at specific scales. Gradients are spatially averaged across channels to produce weighting factors. These weights multiply corresponding activation maps, generating a scale-specific aggregated activation map. The activation maps are normalized, resized, and overlaid onto the original CT slices to visualize key regions influencing model predictions at anatomical and textural scales. Supplementary Figure S1 presents these activation maps for representative CT scans, demonstrating distinct visual patterns associated with anatomical (larger, smoother structures) versus textural scales (finer details). The anatomical scale activation maps emphasize large-scale structures such as organ boundaries, overall tumor shapes, and major anatomical landmarks, validating the effectiveness of PDE-driven diffusion. Conversely, textural scale maps highlight detailed internal textures, small lesions, and vascular patterns, confirming that the PDE-based design captures clinically relevant high-frequency features. This multi-scale visualization underscores SCOPE-MIA's ability to provide interpretable, scale-consistent representations beneficial for clinical decision-making and robust medical image analysis.

## S3: Regression Plots for Predicted PyRadiomics Features Using SCOPE-MIA Embeddings to Predict PyRadiomics Features

We assessed the predictability of 107 extracted features by evaluating the coefficient of determination ($R^2$) from regression models trained to approximate each feature individually. The $R^2$ score provides a measure of how well the variability of a given feature is captured by the model. A score of 1 indicates perfect prediction, while values near zero suggest that the model fails to capture underlying structure.

To characterize predictability, we visualized $R^2$ scores sorted in descending order (Figure S2). A horizontal reference line is included at $R^2 = 0.5$ to demarcate features with high model alignment. The individual regression plots for all 107 features are provided in Supplementary Figures S3-S6. The numeric IDs used in these plots correspond to the feature names listed in Supplementary Table 1.

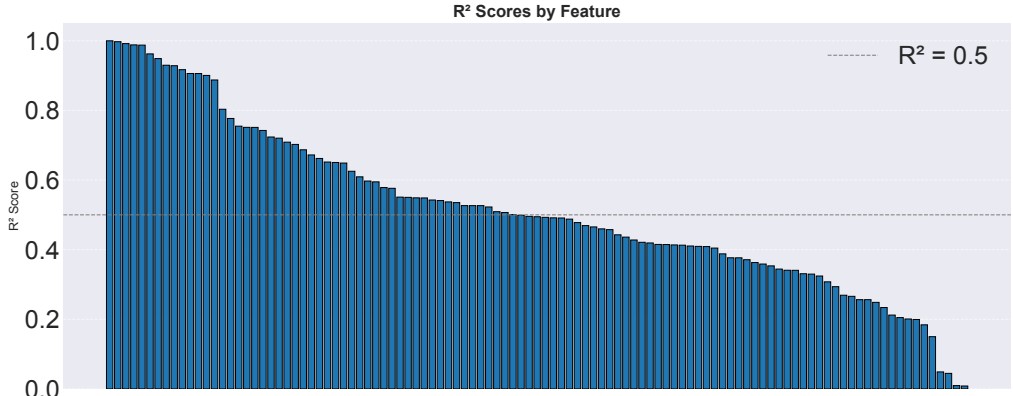

Supplementary Figure S2: $R^2$ scores of all 107 features, sorted in descending order. A horizontal dashed line at $R^2 = 0.5$ highlights the threshold for strong predictability.

Of the 107 features analyzed:

- 51 features exhibited $R^2$ scores above 0.5, indicating strong agreement between the model and the underlying structure of those features.

- 90 features achieved scores above 0.3, suggesting that the majority of features possess moderate to strong predictability.

- Only 7 features had $R^2$ scores below 0.2, reflecting limited model alignment.

Closer inspection reveals that the most predictable features are largely geometric descriptors, such as axis lengths, shape elongation, and diameter measures. These features often encode physical or anatomical structures that are consistent across samples, leading to stronger signal-to-noise ratios and improved model performance. Their high $R^2$ scores support their reliability as derived representations in the context of downstream learning tasks.

In contrast, the small subset of poorly predicted features likely reflects either high intrinsic noise or more complex, nonlinear dependencies that are not well captured by the current modeling approach. These features were retained for completeness but excluded from interpretability-driven analyses due to their low explanatory value.

As a result of this analysis, we focused subsequent modeling and feature interpretation efforts on the 51 highly predictable features ($R^2 > 0.5$). This filtering step allowed us to reduce dimensionality and enhance interpretability while retaining the most informative subset of the feature space. Detailed scatter plots for all 107 PyRadiomics features, illustrating the actual versus predicted values, are shown in Supplementary Figures S3-S6.

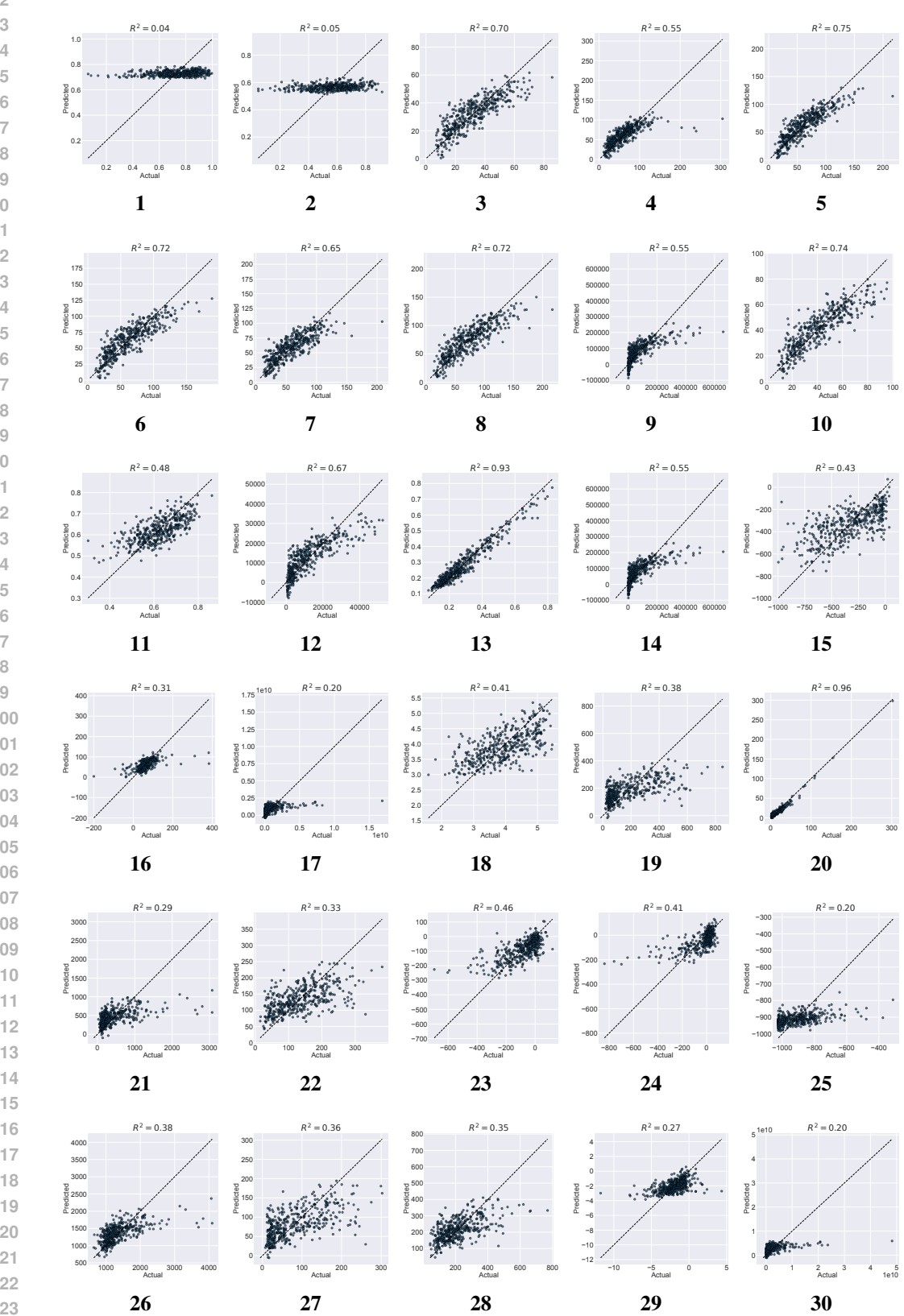

Supplementary Figure S3. Regression scatter plots for targets 1–30.

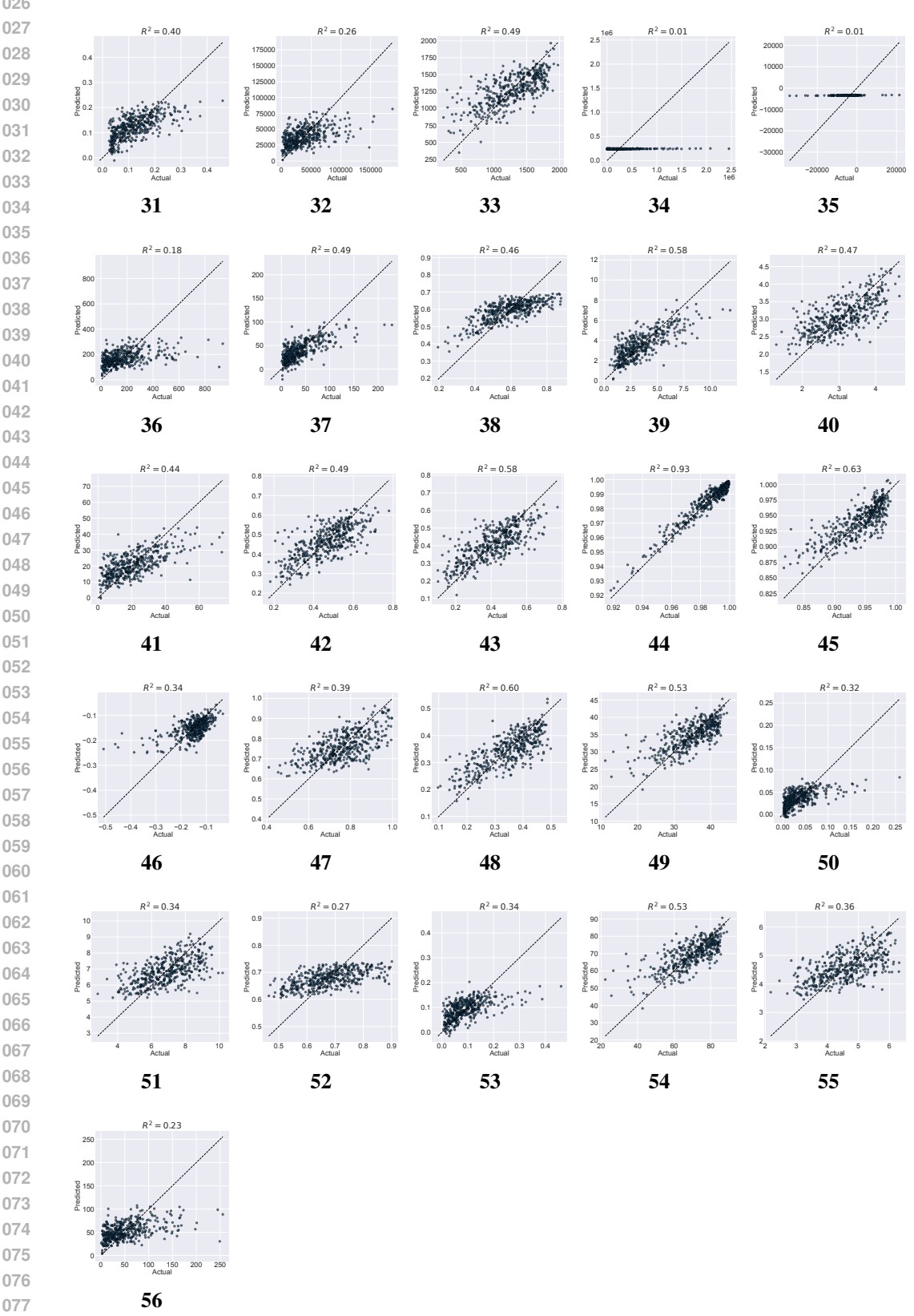

Supplementary Figure S4. Regression scatter plots for targets 31–56.

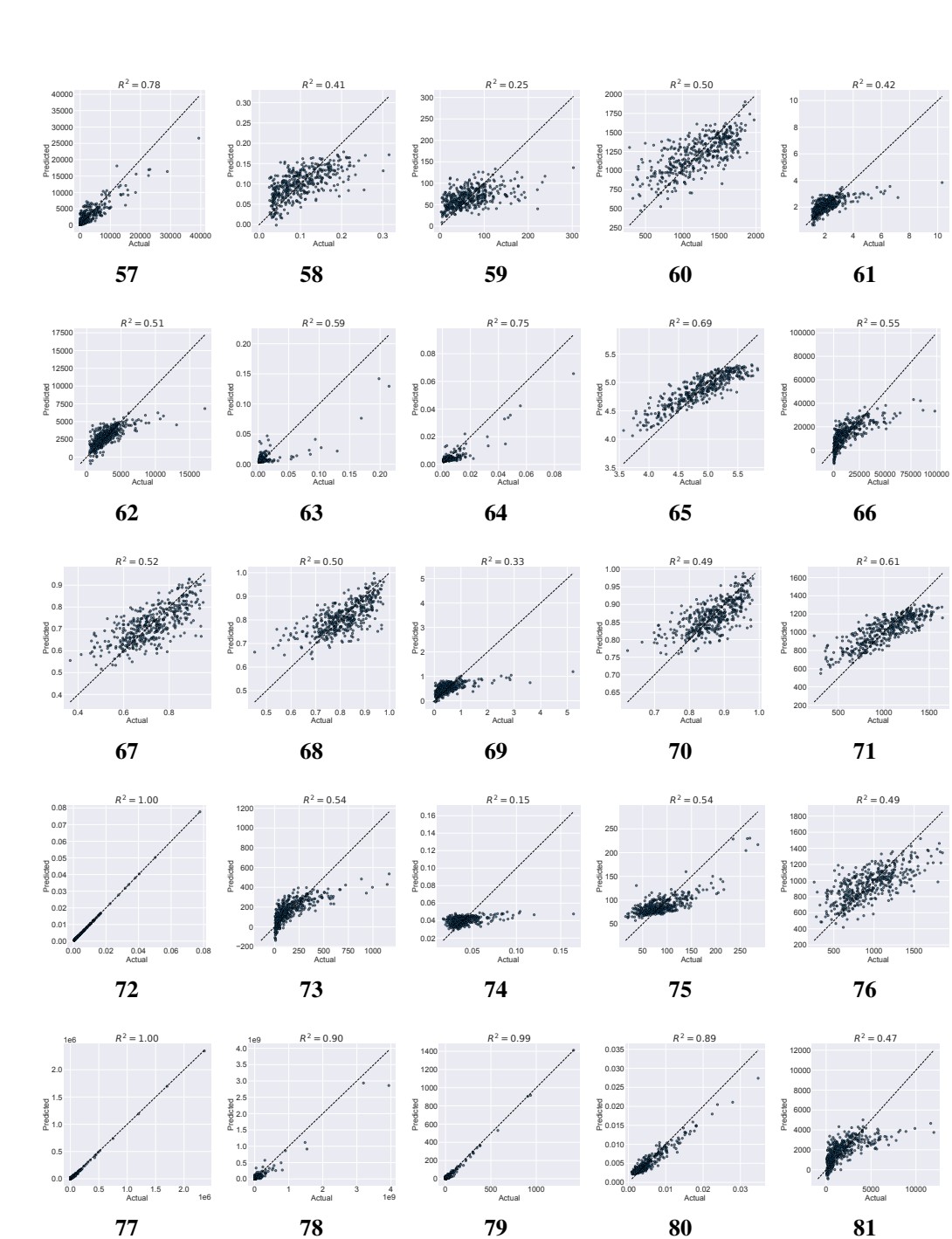

Supplementary Figure S5. Regression scatter plots for targets 57–81.

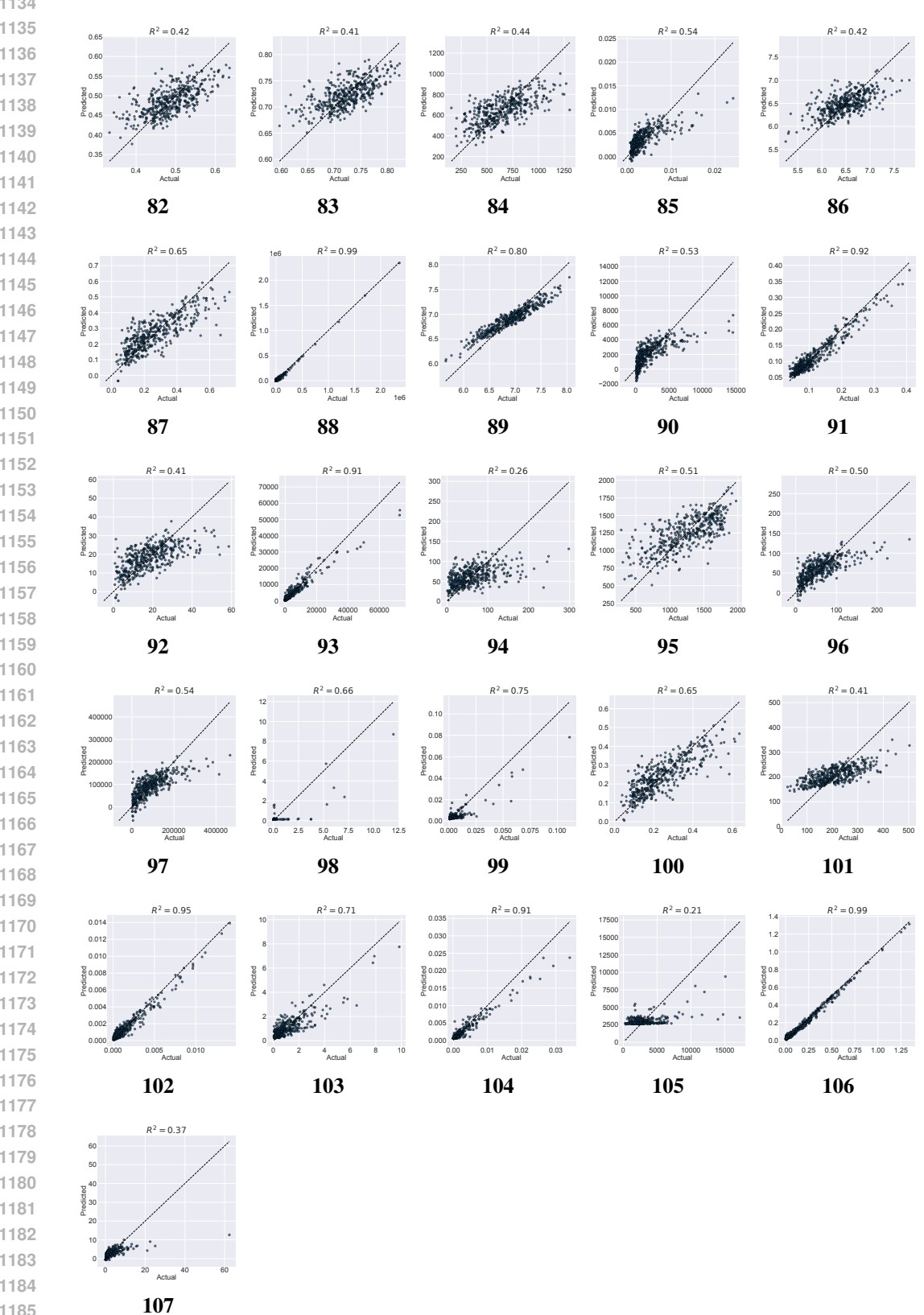

Supplementary Figure S6. Regression scatter plots for targets 82–107.

| ID | Feature Name | ID | Feature Name |
|----|---|---|---|
| 1 | original_shape_Elongation | 55 | original_glcm_SumEntropy |
| 2 | original_shape_Flatness | 56 | original_glcm_SumSquares |
| 3 | original_shape_LeastAxisLength | 57 | original_glrlm_GrayLevelNonUniformity |
| 4 | original_shape_MajorAxisLength | 58 | original_glrlm_GrayLevelNonUniformityNormalized |
| 5 | original_shape_Maximum2DDiameterColumn | 59 | original_glrlm_GrayLevelVariance |
| 6 | original_shape_Maximum2DDiameterRow | 60 | original_glrlm_HighGrayLevelRunEmphasis |
| 7 | original_shape_Maximum2DDiameterSlice | 61 | original_glrlm_LongRunEmphasis |
| 8 | original_shape_Maximum3DDiameter | 62 | original_glrlm_LongRunHighGrayLevelEmphasis |
| 9 | original_shape_MeshVolume | 63 | original_glrlm_LongRunLowGrayLevelEmphasis |
| 10 | original_shape_MinorAxisLength | 64 | original_glrlm_LowGrayLevelRunEmphasis |
| 11 | original_shape_Sphericity | 65 | original_glrlm_RunEntropy |
| 12 | original_shape_SurfaceArea | 66 | original_glrlm_RunLengthNonUniformity |
| 13 | original_shape_SurfaceVolumeRatio | 67 | original_glrlm_RunLengthNonUniformityNormalized |
| 14 | original_shape_VoxelVolume | 68 | original_glrlm_RunPercentage |
| 15 | original_firstorder_10Percentile | 69 | original_glrlm_RunVariance |
| 16 | original_firstorder_90Percentile | 70 | original_glrlm_ShortRunEmphasis |
| 17 | original_firstorder_Energy | 71 | original_glrlm_ShortRunHighGrayLevelEmphasis |
| 18 | original_firstorder_Entropy | 72 | original_glrlm_ShortRunLowGrayLevelEmphasis |
| 19 | original_firstorder_InterquartileRange | 73 | original_glszm_GrayLevelNonUniformity |
| 20 | original_firstorder_Kurtosis | 74 | original_glszm_GrayLevelNonUniformityNormalized |
| 21 | original_firstorder_Maximum | 75 | original_glszm_GrayLevelVariance |
| 22 | original_firstorder_MeanAbsoluteDeviation | 76 | original_glszm_HighGrayLevelZoneEmphasis |
| 23 | original_firstorder_Mean | 77 | original_glszm_LargeAreaEmphasis |
| 24 | original_firstorder_Median | 78 | original_glszm_LargeAreaHighGrayLevelEmphasis |
| 25 | original_firstorder_Minimum | 79 | original_glszm_LargeAreaLowGrayLevelEmphasis |
| 26 | original_firstorder_Range | 80 | original_glszm_LowGrayLevelZoneEmphasis |
| 27 | original_firstorder_RobustMeanAbsoluteDeviation | 81 | original_glszm_SizeZoneNonUniformity |
| 28 | original_firstorder_RootMeanSquared | 82 | original_glszm_SizeZoneNonUniformityNormalized |
| 29 | original_firstorder_Skewness | 83 | original_glszm_SmallAreaEmphasis |
| 30 | original_firstorder_TotalEnergy | 84 | original_glszm_SmallAreaHighGrayLevelEmphasis |
| 31 | original_firstorder_Uniformity | 85 | original_glszm_SmallAreaLowGrayLevelEmphasis |
| 32 | original_firstorder_Variance | 86 | original_glszm_ZoneEntropy |
| 33 | original_glcm_Autocorrelation | 87 | original_glszm_ZonePercentage |
| 34 | original_glcm_ClusterProminence | 88 | original_glszm_ZoneVariance |
| 35 | original_glcm_ClusterShade | 89 | original_gldm_DependenceEntropy |
| 36 | original_glcm_ClusterTendency | 90 | original_gldm_DependenceNonUniformity |
| 37 | original_glcm_Contrast | 91 | original_gldm_DependenceNonUniformityNormalized |
| 38 | original_glcm_Correlation | 92 | original_gldm_DependenceVariance |
| 39 | original_glcm_DifferenceAverage | 93 | original_gldm_GrayLevelNonUniformity |
| 40 | original_glcm_DifferenceEntropy | 94 | original_gldm_GrayLevelVariance |
| 41 | original_glcm_DifferenceVariance | 95 | original_gldm_HighGrayLevelEmphasis |
| 42 | original_glcm_Id | 96 | original_gldm_LargeDependenceEmphasis |
| 43 | original_glcm_Idm | 97 | original_gldm_LargeDependenceHighGrayLevelEmphasis |
| 44 | original_glcm_Idmn | 98 | original_gldm_LargeDependenceLowGrayLevelEmphasis |
| 45 | original_glcm_Idn | 99 | original_gldm_LowGrayLevelEmphasis |
| 46 | original_glcm_Imc1 | 100 | original_gldm_SmallDependenceEmphasis |
| 47 | original_glcm_Imc2 | 101 | original_gldm_SmallDependenceHighGrayLevelEmphasis |
| 48 | original_glcm_InverseVariance | 102 | original_gldm_SmallDependenceLowGrayLevelEmphasis |
| 49 | original_glcm_JointAverage | 103 | original_ngtdm_Busyness |
| 50 | original_glcm_JointEnergy | 104 | original_ngtdm_Coarseness |
| 51 | original_glcm_JointEntropy | 105 | original_ngtdm_Complexity |
| 52 | original_glcm_MCC | 106 | original_ngtdm_Contrast |
| 53 | original_glcm_MaximumProbability | 107 | original_ngtdm_Strength |
| 54 | original_glcm_SumAverage | | |

Supplementary Table 1: Feature mapping: Numeric IDs correspond to the PyRadiomics feature names. The left two columns list features 1–54, and the right two columns list features 55–107 (the final cell is left blank).

