# OpenReview forum: "SCOPE-MIA: Scale-Consistent Partial Differential Equation-Optimized Encoding in 3D Medical Imaging Analysis"
_ICLR.cc/2026/Conference — ICLR 2026 Conference Withdrawn Submission_

### Official Review · Reviewer_gMPZ · 2025-10-28

**Soundness:** 2
**Presentation:** 3
**Contribution:** 3
**Rating:** 4
**Confidence:** 4

**Summary:**

This paper introduces SCOPE-MIA, a novel framework for the analysis of 3D medical images.
Core contributions:
1.  Theoretical Innovation: It pioneers the integration of tailored Partial Differential Equations (PDEs) as an internal constraint, achieving the explicit disentanglement of anatomical and textural features, establishing a new paradigm for medical image analysis.
2.  Architectural Design: It introduces plug-and-play "PDE Blocks" integrated into modern 3D CNNs, enabling principled decomposition and refinement of multi-scale features within the latent space via dual-pathway PDE dynamics.
3.  Mechanism Innovation: It incorporates entropy-adaptive diffusion and information-theoretic attention mechanisms into the PDE processes, allowing the model to intelligently adapt feature processing based on local image content (e.g., tumor margins) for enhanced robustness.
4.  Empirical Breakthrough: Comprehensive experiments demonstrate that the framework significantly outperforms the clinical gold-standard of handcrafted radiomics across diverse cancer imaging tasks, providing a more powerful tool for image-based precision medicine.

**Strengths:**

1.  Novel and Well-Motivated Conceptual Foundation: The core idea of embedding tailored Partial Differential Equation (PDE) constraints to explicitly disentangle anatomical and textural features within a deep learning framework is highly innovative. It provides a principled inductive bias that directly addresses the feature entanglement problem in standard CNNs.
2.  Rigorous and Clinically Relevant Benchmarking: The choice of the IBSI-compliant PyRadiomics pipeline as the primary benchmark is a significant strength. Demonstrating clear superiority over this established clinical gold-standard is a more impactful and meaningful validation than comparisons against other deep learning models alone, as it directly bridges the gap between advanced research and clinical practice.
3.  Comprehensive Experimental Validation: The framework is thoroughly evaluated across a wide range of conditions, including multiple public datasets (NSCLC, TCGA, BCBM), imaging modalities (CT, MRI), and clinically critical endpoints (e.g., EGFR status, survival, age, cancer stage). This demonstrates the generalizability and robustness of the proposed method.
4.  In-Depth Analysis and Ablation: The paper goes beyond simple performance reporting. The inclusion of a detailed ablation study (Table 1) convincingly demonstrates the contribution of each proposed component (e.g., entropy adaptation, attention guidance). Furthermore, the computational complexity analysis (Tables 4 & 5) provides a transparent assessment of the method's practical overhead.

**Weaknesses:**

1.  Insufficient Comparison to Deep Learning Baselines: The paper lacks direct and fair performance comparisons against state-of-the-art 3D deep learning models (e.g., plain 3D ResNet, ConvNeXt, or Vision Transformers). The primary benchmark against PyRadiomics alone makes it difficult to discern whether the performance gains stem from the novel PDE components or merely from the use of a powerful 3D CNN backbone.
2.  Vagueness in Methodological Details: Several critical implementation details of the proposed "PDE Blocks" remain unclear. Key aspects, such as which PDE parameters (e.g., diffusion coefficients) are fixed versus learnable, and the precise mechanism for splitting features into anatomical and textual streams (predefined or adaptive), are not sufficiently specified, hindering reproducibility.
3.  Inadequate Evidence for Feature Disentanglement: The central claim of achieving disentangled anatomical and textural features is not robustly supported. The evidence relies heavily on qualitative activation maps and speculative descriptions of the latent space, lacking quantitative metrics (e.g., mutual information) or direct experiments (e.g., latent space manipulation) to rigorously demonstrate the independence and semantic purity of the two feature streams.
4.  Incomplete Justification for Computational Overhead: While the paper acknowledges the computational overhead introduced by the PDE modules, it does not sufficiently justify this cost. There is no efficiency-versus-performance trade-off analysis to demonstrate that the performance gain is indeed superior to simply using a larger standard model under a similar computational budget.

**Questions:**

1.  Are key parameters in the PDE Blocks (e.g., diffusion coefficients) fixed or learnable, and how are they implemented?
2.  Why not compare against the base network without PDE Blocks to prove the performance gain truly stems from the PDE components?
3.  Beyond qualitative visualizations, are there quantitative metrics (e.g., mutual information) to directly validate the effectiveness of feature disentanglement?

---

### Official Review · Reviewer_sRU7 · 2025-10-31

**Soundness:** 3
**Presentation:** 3
**Contribution:** 3
**Rating:** 4
**Confidence:** 3

**Summary:**

This paper introduces SCOPE-MIA, a 3D CNN framework that integrates Partial Differential Equations (PDEs) constraints that explicitly
decomposes learned features into distinct anatomical and textural components.The method embeds tailored PDE constraints into a modern 3D CNN architecture to promote the learning of robust, scale-explicit, and disentangled representations. The method is validated against handcrafted features across different cancer imaging datasets and tasks (EGFR mutation, prediction, and survival).

**Strengths:**

1. Theoritically sound approach: the use of PDE constraints to strenghten anatomical and textural decomposition is interesting and relevant.
2. Methodology. The methodology is clear and well defined. Considerations for GPU memory requirements are taken into account with the subvolume partitioning approach.
3. Extended ablation studies. All components are ablated in Table 1 i.e. PDE stability loss, entropy adaptation, Hessian regularization to give an account of the relevance of each of this components in the performance.
4. Comprehensive validation: different datasets (NSCLC-Radiogenomics, TCGA-LUAD/LUSC, BCBM-RadioGenomics) are used to validate the method as well as multiple modalities (CT, MRI). This reinforces the generalisation capabilities of the current approach.

**Weaknesses:**

1. Limited baseline comparisons: the only baseline comparison is against handcrafted PyRadiomics features. There is no comparison with other deep learning methods.

2. Limited performance improvements: Figure 5, the age prediction R square improves from 0.52 to 0.85 but for other tasks the performance gains appears limited. Considering the increased computational requirements (Table 5), some improvements in performance might not be significant enough to justify the additional costs.

3. Implementation details missing: Implementation details are provided in appendix but some critical details are missing i.e.
- Information on how the diffusion coefficients in the PDE Block are initialised and how they are learnable is not provided.
- Information on hyperparameter selection and sensitivity analysis for the lambda parameters.

4. Clinical Validation: there is no insights into the significance of the clinical thresholds. How is AUC transferable clinically? There is also no breakdown of the performance per cancer subtype.

**Questions:**

1. The paper would be strenghtened by including comparison with recent deep learning baselines. Could you please provide 3D ResNet50 and ConvNeXT comparisons with no PDE blocks keeping the same experimental setup?

2. How are the diffusion coefficients  D_A and D_T learnt ? How is performance sensitive to the choice of PDE hyperparameters? This is missing from ablation studies in Table 1.

3. What is the training time compared to baselines?

4. How did you select the PyRadiomics configuration settings?

---

### Official Review · Reviewer_8CTG · 2025-11-01

**Soundness:** 2
**Presentation:** 2
**Contribution:** 1
**Rating:** 2
**Confidence:** 4

**Summary:**

The authors proposed SCOPE-MIA. The main goal is to enhance interpretability and robustness of 3D medical-image representations by embedding Partial Differential Equation (PDE) constraints into 3D CNN. The authors mentioned that by splitting into low-frequency (anatomical) and high-frequency (textural) pathways, the embeddings could be better and thus downstream model performance could be improved. However, many major weaknesses exist, which could be seen below.

**Strengths:**

1. Easy read, good writing.
2. The concept is relatively novel - embedding PDE physics inside a CNN to enforce feature disentanglement.
3. The assumptions are relatively valid, in comparison to radiomics.

**Weaknesses:**

1. Only one commonly used measurement (SSIM) is used to evaluate the performance.
2. Not clear definition of one of the major measurements (Anatomical Part Reconstruction Score (R))
3. Baseline comparisons is highly insufficient. The authors only compare their proposal with ONE baseline model (PyRadiomics), which is a non-DL method. NO DL baselines have been compared with.
4. The authors stated the methods is having scope to “medical imaging” in general, but the evaluation was conducted only on one scenario: predicting EGFR mutation status.

**Questions:**

1. How was the “Anatomical Part Reconstruction Score (R)” calculated? Do the authors have any references, or any justifications about the formula?
2.  “We evaluated both models on the NSCLC-Radiogenomics, TCGA-LUAD & TCGA-LUSC datasets” what are the specs of these datasets? The readers might not be familiar with these datasets, so probably a brief introduction is better.

---

### Official Review · Reviewer_WiBn · 2025-11-01

**Soundness:** 2
**Presentation:** 2
**Contribution:** 2
**Rating:** 2
**Confidence:** 3

**Summary:**

This paper proposes SCOPE-MIA, a 3D deep learning framework that integrates Partial Differential Equation (PDE) constraints into modern convolutional architectures for medical image analysis. The motivation is that existing 3D CNNs learn entangled structural and textural features, mixing smooth anatomical shapes and high-frequency textures within the same feature maps, which can reduce interpretability and robustness. To address this, SCOPE-MIA explicitly decomposes the latent feature space into two complementary streams:
- An anatomical stream governed by a structure-enhancing diffusion PDE that smooths low-frequency components (large-scale organ and lesion structures).
- A textural stream governed by a detail-preserving PDE that retains or amplifies high-frequency information (fine tissue textures).

**Strengths:**

1. Strong conceptual motivation and clinical relevance
- Tackles a fundamental limitation of medical deep learning: lack of interpretability and multi-scale separation between structural (organ-level) and textural (tissue-level) signals.
2. Empirical robustness and generalization
- Demonstrates consistent improvement across multiple tumor sites, imaging modalities (CT and MRI), and prediction tasks (classification, regression).

**Weaknesses:**

1. Limited interpretability on the z embeddings
- Although the latest representations is decomposed into 2 different representations and these 2 representations are further transformed with the mathematical theory, it is difficult to ensure that the representations is corresponding to either textual or anatomical. Also, how can we know the entropy-based coefficients is the correct inductive bias to add on the latent representation? Too many loss functions are controlling the model convergence, leading to a hard time for me to determine why the output is better. It will be great if you can show more visualizations about the embedding distributions, instead of looking into the Grad-CAM, because the Grad-CAM only provides information that how the model's gradient flow through.

2. Weird definition on anatomical and textural details
- What is the difference between anatomical and textural? Normally we will separate contrast and texture features, learning it independently. Can you clarify more on this?

3. The core novelty of this paper
- After reading the whole paper, although the core innovation is to adaptive mathematical inductive bias into the latent representation, in fact, the core idea should be proposed a better representations for downstream tasks. Therefore, I cannot determine whether the gain in Table 3 is due to the better capability of the network architecture itself, or your proposed method. Can you provide more clarity on this?

Minor:
1. ConvNeXt is not a 3D backbone, it is a completely 2D backbone. If the author wants to use the 3D version ConvNeXt as the backbone, please either leverage or cite 3D UX-Net as your baseline method
2. Line 142 has missed citation for Swin-UNETR

**Questions:**

All questions are proposed in the weakness section, please take a look.

---

### Note · Authors · 2025-11-14

I have read and agree with the venue's withdrawal policy on behalf of myself and my co-authors.